

# Deep learning for constructing microblog behavior representation to identify social media user's personality

Xiaoqian Liu and Tingshao Zhu

Institute of Psychology, Chinese Academy of Sciences, Beijing, China

## ABSTRACT

Due to the rapid development of information technology, the Internet has gradually become a part of everyday life. People would like to communicate with friends to share their opinions on social networks. The diverse behavior on socials networks is an ideal reflection of users' personality traits. Existing behavior analysis methods for personality prediction mostly extract behavior attributes with heuristic analysis. Although they work fairly well, they are hard to extend and maintain. In this paper, we utilize a deep learning algorithm to build a feature learning model for personality prediction, which could perform an unsupervised extraction of the Linguistic Representation Feature Vector (LRFV) activity without supervision from text actively published on the Sina microblog. Compared with other feature extractsion methods, LRFV, as an abstract representation of microblog content, could describe a user's semantic information more objectively and comprehensively. In the experiments, the personality prediction model is built using a linear regression algorithm, and different attributes obtained through different feature extraction methods are taken as input of the prediction model, respectively. The results show that LRFV performs better in microblog behavior descriptions, and improves the performance of the personality prediction model.

## INTRODUCTION

Personality can be defined as a set of traits in behaviour, cognition and emotion which is distinctive among people (*Mischel, Shoda & Ayduk, 2007*). In recent years, researchers have formed a consensus on personality structure, and proposed the Big Five factor model (*Costa & McCrae, 1992*), which uses five broad domains or factors to describe the human personality, including openness (O), conscientiousness (C), extraversion (E), agreeableness (A) and neuroticism (N) (*Funder, 2001*).

Traditionally, questionnaires have been widely used for personality assessment, especially the Big Five personality questionnaire. But the form of questionnaire may be inefficient on large population. Due to the rapid development of information technology, the Internet has become part of everyday life. People prefer expressing their thoughts and interacting with friends on social network platforms. Therefore, researchers pay more and more attention to figuring out the correlation between the behavior of users on social networks and their personality traits in order to realize automatic personality prediction by machine learning methods.

Corresponding authors
Xiaoqian Liu, liuxiao-qian@psych.ac.cn
Tingshao Zhu, tszhu@psych.ac.cn

Nowadays, the Internet is not used just for communication, but also as a platform for users to express their thoughts, ideas and feelings. Personality is indirectly expressed by users' behavior on the social network, which refers to a variety of operations on the social network, such as comment, follow and like. In addition, text, punctuation and emoticons published by users can be regarded as one kind of social behavior. Therefore, for automatic personality prediction, how to abstract these diverse and complex behaviors and acquire the digital representation of social network behaviors has become an critic problem. Existing behavior analysis methods are mostly based on some statistics rules, but artificial means have some disadvantages in objectivity and integrity. Generally, attributes are especially important for the performance of a prediction model. A set of proper feature vectors could improve the effectiveness of prediction model to a certain extent. Therefore, it is required that the attributes are not only the comprehensive and abstract description of individual's behavior characteristic, but also reflect the diversity of different individuals' behaviors.

In this paper, we use a deep learning algorithm to perform an unsupervised extraction LRFV from users' content published on the Sina microblog. Compared with other attributes obtained by artificially means, LRFV could represent users' linguistic behavior more objectively and comprehensively. There are two reasons of utilizing deep learning algorithm to investigate the correlation between users' linguistic behavior on social media and their personality traits. One is that deep learning algorithm could extract high-level abstract representation of data layer by layer by exploiting arithmetical operation and the architecture of model. It has been successfully applied in computer vision, object recognition and other research regions. Another is that the scale of social network data is huge, and the deep learning algorithm can meet the computational demands of big data. Given all this, in this article we have done a preliminary study on constructing microblog linguistic representation for personality prediction based on the deep learning algorithm.

## Related work

At present, many researchers have paid attention to the correlation between users' Internet behaviors and their personality traits. *Qiu et al. (2012)* investigated the relationship between tweets delivered on Twitter and users' personality, and they found that some personality characteristics such as openness (O), extraversion (E) and agreeableness (A) are related to specific words used in tweets. Similarly, *Vazire & Gosling (2004)* discovered that there is great relevance between users' specific Internet behaviors and their personality through studying users' behaviors on personal websites. These conclusions can be explained as personality not only influencing people's daily behaviors, but also playing an important role in users' Internet behaviors. With the rise of social media, more and more researchers begin to analyse users' personality traits through social network data with the help of computer technology. *Sibel & Golbeck (2014)* predicted users' personality based on operational behaviors on Twitter utilizing linear regression model. Similarly, *Golbeck et al. (2011)* used regression algorithm to build a personality prediction model, but they considered both of operational behaviors and linguistic behaviors. *Lima & De Castro (2013)* used a semi-supervised method to predict personality based on the attributes of linguistic behaviors extracted from tweets. *Ortigosa, Carro & Quiroga (2013)* built a personality prediction

model of users according to their social interactions in Facebook by machine-learning methods, such as classification trees.

Although many researchers utilized machine learning methods to built personality prediction models and have made some achievements, there are also some disadvantages for which improvements are needed. First, in state-of-art methods, the behavior analysis method and behavior attributes extraction methods are mostly based on some experiential heuristic rules which are set artificially. The behavior attributes extracted manually by statistical methods may not be able to comprehensively and objectively describe the characteristics of behaviors. Second, supervised and semi-supervised behavior feature extraction methods need a certain amount of labeled data, but obtaining a large number of labeled data is difficult, time-consuming and has a high cost. Therefore, supervised and semi-supervised feature extraction methods are not suitable for a wide range of application.

### Deep learning

In recent years, there are more and more interdisciplinary research of computational science and psychology (*Zhang et al., 2013*; *Chen et al., 2015*). Deep learning is a set of algorithms in machine learning (*Bengio, 2009*; *Bengio, Courville & Vincent, 2013*), which owns a hierarchical structure in accordance with the biological characteristics of human brain. The deep learning algorithm originated in artificial neural networks, and has been applied successfully in many artificial intelligence applications, such as face recognition (*Huang, Lee & Learned-miller, 2012*), image classification (*Ciresan, Meier & Schmidhuber, 2012*), natural language processing (*Socher et al., 2013*) and so on. Recently, researchers are attempting to apply deep learning algorithms to other research fields. *Huijie et al. (2014a)* and *Huijie et al. (2014b)* used the Cross-media Auto-Encoder (CAE) to extract feature vectors, and identified users' psychological stress based on social network data. Due to the multi-layer structure and mathematics algorithm design, deep learning algorithms can extract more abstract high-level representation from low-level feature through multiple non-linear transformations, and discover the distribution characteristics of data. In this paper, based on the deep learning algorithm, we could train unsupervised linguistic behavior feature learning models for five factors of personality. Through the feature learning models, the LRFV corresponding to each personality trait can be learned actively from users' contents published on the Sina microblog. The LRFV could describe the users' linguistic behavior more objectively and comprehensively, and improve the accuracy of the personality prediction model.

## DATASET

In this paper, we utilize deep learning algorithm to construct an unsupervised feature learning model which can actively and objectively extract the Linguistic Representation Feature Vector (LRFV) from users' content published on the Sina microblog. Next, five personality prediction models corresponding to five personality traits are built using a linear regression algorithm based on LRFV. We conducted preliminary experiments on relatively small data as pre-study of exploring the feasibility of using deep learning algorithm to investigate the correlation between a user's social network behavior and personality.

## Data collection

Nowadays, users prefer expressing their attitudes and feelings through social network. Therefore, the linguistic information on social network is more significant for analysing users' personality characteristics. In this paper, we pay more attention to the correlation between users' linguistic behaviors on Sina microblog and their personalities. According to the latest statistics, by the end of Dec. 2014, the total number of registered users of Sina microblog has exceeded 500 million. On the 2015 spring festival's eve, the number of daily active users was more than one billion. It can be said that the Sina microblog is currently one of the most popular social network platforms in China. Similarly to Facebook and Twitter, Sina microblog users can post blogs to share what they saw and heard. Through the Sina microblog, people express their inner thoughts and ideas, follow friends or someone they want to pay attention to, and comment or repost blogs that interest them or on which they agree.

For data collection, we firstly released the experiment recruitment information on Sina microblog. Based on the assumption that the users are often express themselves on social media platform, we try to construct personality prediction model. So, it is required that for one person, there have to be enough Sina microblog data. On the other hand, some participants might provide their deprecated or deputy accounts of social network rather than the commonly used and actual accounts when participating our experiment; such data are unfaithful. In consideration of this, we set an "active users" selection criteria for choosing the effective and authentic samples.

Our human study has been reviewed and approved by the Institutional Review Board, and the protocol number is "H09036." In totally, 2,385 volunteers were recruited to participate in our experiments. They have to finish the Big Five questionnaire (*Vittorio et al., 1993*) online and authorized us to obtain the public personal information and all blogs. Collecting volunteers' IDs of Sina microblog, we obtained their microblog data through Sina microblog API. The microblog data collected consists of the users' all blogs and their basic status information, such as age, gender, province, personal description and so on. The whole process of subjects recruitment and data collection lasted nearly two months. Through the preliminary screening, we obtained 1,552 valid samples finally. When filtering invalid and noisy data, we designed some heuristic rules as follows:

- If the total number of one's microblogs is more than 500, this volunteer is a valid sample. This rule can ensure that the volunteer is an active user.
- In order to ensure the authenticity of the results of questionnaire, we set several polygraph questions in the questionnaire. The samples with unqualified questionnaires were removed.
- When the volunteers filled out the questionnaire online, the time they took on each question were recorded. If the answering time was too short, the corresponding volunteer was considered as an invalid sample. In our experiments, we set that the answering time should be longer than 2 s.

## Data for linguistic behavior feature learning

Through iteration and calculation layer by layer, a deep learning algorithm can mine the internal connection and intrinsical characteristics of linguistic information on social network platforms. Assuming that the text in microblogs could reflect users' personality characteristics, for each trait of personality, we build a linguistic behavior feature learning model based on the deep learning algorithm to extract the corresponding LRFV from users' expressions in the Sina microblog.

Linguistic Inquiry and Word Count (LIWC) is a kind of language statistical analysis software, which has been widely used by many researchers to extract attributes of English contents from Twitter and Facebook (*Golbeck et al., 2011*; *Golbeck, Robles & Turner, 2011*). In order to meet the demands of simple Chinese semantic analysis, we developed a simplified Chinese psychological linguistic analysis dictionary for the Sina microblog (SCLIWC) (*Gao et al., 2013*). This dictionary was built based on LIWC 2007 (*Tausczik & Pennebaker, 2010*) and the traditional Chinese version of LIWC (CLIWC) (*Huang et al., 2012*). Besides referring to the original LIWC, we added five thousand words which are most frequently used in the Sina microblog into this dictionary. The words in the dictionary are classified into 88 categories according to emotion and meaning, such as positive word, negative word, family, money, punctuation, etc. Through analysis and observation, we found that in some factors of personality, users of different scores show great differences in the number of used words belonging to positive emotion, negative emotion and some other categories in the dictionary.

According to SCLIWC (*Gao et al., 2013*), the users' usage degree of words in blogs could be computed in 88 categories. In order to obtain the usage characteristics of social media text in the temporal domain, we first divide the time by week. For the $i$th word category of SCLIWC, the usage frequency within the $j$th week $f_j^i$ ($i = 1, 2, \ldots, 88$) is calculated by Eq. (1), in which, $i$ denotes the serial number of category, and $j$ denotes the serial number of week. We collect all the text published in Sina microblog during recent three years (Jun. 2012–Jun. 2015), and there are 156 weeks in total. Therefore, corresponding to each category of SCLIWC, the vector $f^i = \{f_1^i, f_2^i, \ldots, f_{156}^i\}$ is the digital representation of the $i$th category in temporal domain.

$$f_j^i = \frac{\text{The number of words belongs to the } i\text{th category of SCLIWC in } j\text{th week}}{\text{The total number of words in blogs in } j\text{th week}}. \tag{1}$$

Then, we utilize Fast Fourier Transform(FFT) (*Loan, 1992*) to obtain the varying characteristics of social media text usage in temporal space. Fourier Transform is a special integral transform, which could convert the original temporal signal into frequency domain signal which is easily analyzed. FFT is the fast algorithm of Discrete Fourier Transform (DFT), defined by

$$X(k) = \text{DFT}[x(n)] = \sum_{n=0}^{N-1} x(n) W_N^{kn}, \quad k = 0, 1, \ldots, N-1 \tag{2}$$

$$W_N = e^{-j\frac{2\pi}{N}}. \tag{3}$$

In order to extract the temporal information from massive high-dimensional digital vectors, Fourier time-series analysis is considered. Concretely, we conduct FFT for each vector. Through FFT, the amplitudes calculated include frequency information, and former 8 maximum amplitudes are selected to constitute a vector as the representation of each word category. Finally, linking the vectors of each category in series, we can obtained a linguistic vector of 704 length corresponding to each user ID.

In our experiment, we use 1,552 users' blogs published in three years as data for preliminary study. Each user's linguistic behavior is represented as vector form through FFT based on SCLIWC.

## Data for personality prediction

In order to verify that the deep learning algorithm is an effective method for extracting a representation of a user's Sina microblog linguistic behaviors, we built a personality prediction model based on linguistic behavior feature vectors. The personality prediction model is constructed by a linear regression algorithm. For each volunteer, five linguistic behavior feature vectors corresponding to five traits of personality are obtained by feature learning models, respectively. The training process of the personality prediction model is supervised, so users' five scores of five personality traits in the Big Five questionnaire are taken as their labels of the corresponding linguistic behavior feature vectors.

## METHODS

### Unsupervised feature learning based on Stacked Autoencoders

Feature learning can be seen as a process of dimensionality reduction. In order to improve the computational efficiency, for all traits of personality, we utilize the relatively simpler form of artificial neural network, autoencoder (*Bengio, 2009*). Figure 1 shows the basic structure of an autoencoder. Basically, for an autoencoder, the input and output own the same dimensions, both of them can be taken as $X$ but, through mathematical transformation, the input and output may be not completely equal. In Fig. 1, $X$ denotes input and $X'$ denotes output. The variable in hidden layer $Y$ is encoded through $X$ by Eq. (4).

$$Y = f_\theta(X) = s(W^T X + b) = s\left(\sum_{i=1}^{n} W_i x_i + b\right) \tag{4}$$

$$s(z) = \frac{1}{1 + exp(-z)}. \tag{5}$$

In Eq. (4), $\{W, b\}$ are parameters which can be obtained through training. $s(z)$ is the Sigmoid activation function of hidden layers which is defined in Eq. (5). In addition, a reconstructed vector $X'$ in input vector space could be obtained by mapping the result of hidden layer $Y$ back through a mapping function,

$$X' = g_{\theta'}(Y) = s'(W'^T Y + b') = s\left(\sum_{i=1}^{n} W_i' y_i + b'\right). \tag{6}$$

For an autoencoder, if we want the mapping result $Y$ as another representation of input $X$, it is assumed that the input $X$ and the reconstructed $X'$ are the same. According to this

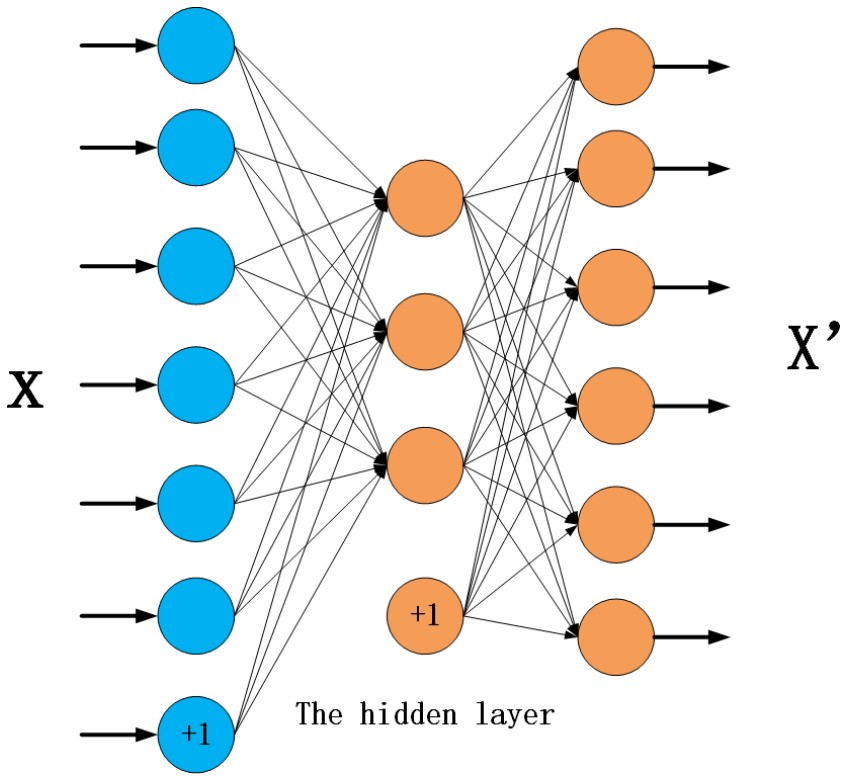

**Figure 1** **The basic structure of an autoencoder.**

assumption, the training process of an autoencoder could be conducted and the parameters of the autoencoder are adjusted according to minimize the error value $L$ between $X$ and $X'$, as shown in the Eq. (7) and Fig. 2. Due to the fact that the error is directly computed based on the comparison between the original input and the reconstruction obtained, the whole training process is unsupervised.

$$L(X; W, b) = \| X' - X \|^2. \tag{7}$$

Several autoencoders are stacked to initialize the deep architectures layer by layer as Fig. 3. Let the hidden layer of $k$th layer be used as the input of $(k+1)$th layer. We used greedy layer-wise training to obtain the optimal parameters $\{W, b\}$ for a Stacked Autoencoder model. That is, the parameters of each layer are trained individually while freezing parameters for the remainder of the model. The output of the $n$th layer $Y^n$ is used as the input of the subsequent $(n+1)$th layer to trained the parameters. The number of layer would be decided according to the optimal value of many experiments. Adjusting the number of layers, and the number of layer corresponding the better performance of prediction model would be set as the optimal number of layer. Then, we take the output of the last layer as the abstract representation of the original linguistic behavior information. Figure 3 shows the structure of our model. For different personality factors, the number of layers and the number of units in each layer are different. The details are presented on the left of Fig. 3. For "A," "C" and "N," there are one hidden layers in the SAE, and the

**Peer**J Computer Science

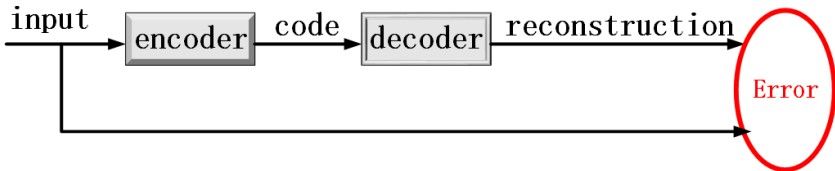

**Figure 2**  The training principle diagram of an autoencoder.

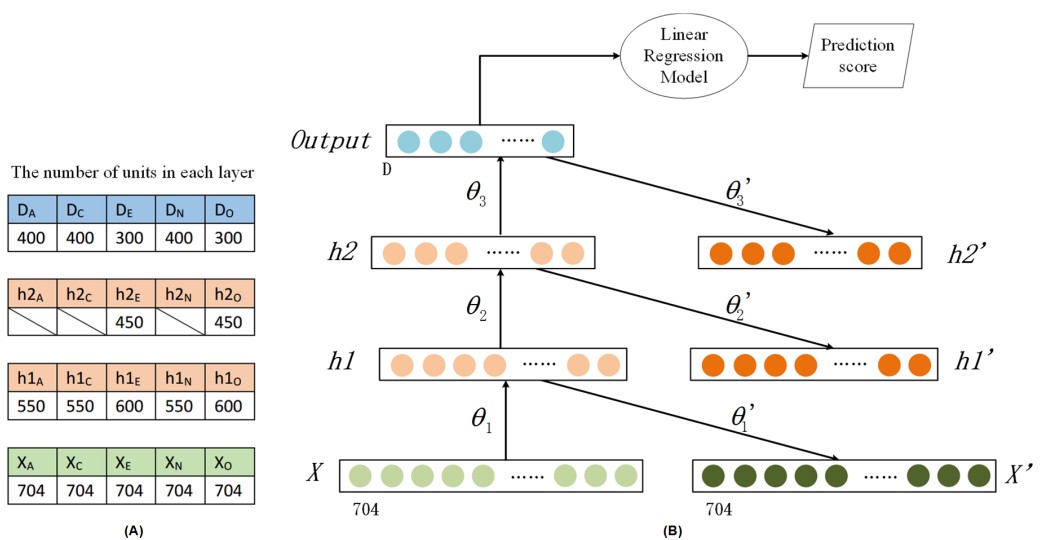

The number of units in each layer

| $D_A$ | $D_C$ | $D_E$ | $D_N$ | $D_O$ |
|-----|-----|-----|-----|-----|
| 400 | 400 | 300 | 400 | 300 |

| $h2_A$ | $h2_C$ | $h2_E$ | $h2_N$ | $h2_O$ |
|-----|-----|-----|-----|-----|
|  |  | 450 |  | 450 |

| $h1_A$ | $h1_C$ | $h1_E$ | $h1_N$ | $h1_O$ |
|-----|-----|-----|-----|-----|
| 550 | 550 | 600 | 550 | 600 |

| $X_A$ | $X_C$ | $X_E$ | $X_N$ | $X_O$ |
|-----|-----|-----|-----|-----|
| 704 | 704 | 704 | 704 | 704 |

(A)

(B)

**Figure 3**  **The deep structure of our prediction model.** The left table shows the details of SAE of the different personality factors.

feature learning model are three layers in total. For "E" and "O," there are two hidden layers in the SAE. In our experiments, 1,552 users' content information of Sina microblog are used as training dataset, and the unsupervised feature learning models corresponding different personality traits are trained respectively. That is, we could obtain five feature learning models in total. For each trait, there will be corresponding linguistic behavior feature vectors extracted from social network behavior data actively.

Finally, based on the Big Five questionnaire, for each user, we could obtained five scores $(S_A, S_C, S_E, S_N, S_O)$ corresponding to "A," "C," "E," "N," "O" five factors, respectively. These scores are used to label corresponding linguistic behavior feature vectors for personality prediction models.

## Personality prediction model based on linear regression

Personality prediction is a supervised process. The linguistic behavior feature vectors are labeled by the corresponding scores of the Big Five questionnaire. For five traits of personality, we utilized the linear regression algorithm to build five personality prediction models in totally.

Take one trait of personality as an example, the linguistic behavior feature vectors are represented by

$$X = \{X_i | X_i = (x_{i1}, x_{i2}, \ldots, x_{im})\}_{i=1}^n, \tag{8}$$

in which, $n$ is the number of samples, $n = 1{,}552$, and $m$ denotes the number of dimensions of the input vector. The scores of the Big Five questionnaire are taken as the labels,

$$Y = \{y_i\}_{i=1}^{n}. \tag{9}$$

The general form of linear regression is

$$y_i = \omega_1 x_{i1} + \omega_2 x_{i2} + \cdots + \omega_m x_{im} + \varepsilon_i, (i = 1, 2, \ldots, n). \tag{10}$$

We trained five personality prediction models based on linear regression algorithm using corresponding linguistic behavior feature vectors and labels.

## RESULTS

In Experiments, we collect 1,552 users' Sina microblog data in total. The linguistic behavior of users are quantified based on SCLIWC, and the temporal characteristics are calculated through FFT. Then, we utilize deep learning algorithm to construct feature learning models, which could extract objective and comprehensive representation of linguistic behaviors from the temporal sequence. Finally, personality prediction model is trained by the linear regression algorithm based on linguistic behavior feature vectors.

### Evaluation measures

In this paper, we conducted preliminary study about constructing microblog behavior representation for predicting social media user's personality. The five factors of personality are all tested. We use the Pearson product-moment correlation coefficient ($r$) and Root Mean Square Error ($RMSE$) to measure the quality of different behavior feature representation methods. The computational formulas of two measurements are shown in Eqs. (11) and (12) respectively. In Eq. (11), $Cov(Y, Y')$ denotes the covariance of $Y$ and $Y'$, and $Var(Y)$ and $Var(Y')$ represents the variances of the real score $Y$ and prediction score $Y'$ respectively. When $r > 0$, it means the results of questionnaire and prediction model have a positive correlation. In contrast, $r < 0$ means negative correlation. The absolute value is greater, the higher is the degree of correlation. In psychology research, we use Cohen's conventions (*Cohen, 1988*) to interpret the Pearson product-moment correlation coefficient. $r \in [0.1, 0.3)$ represent a weak or small association and $r \in [0.3, 0.5)$ indicates a moderate correlation between two variables. In Eq. (12), $i$ is the sequence number of sample and $n$ is the total number of samples, $n = 1{,}552$. In the Big Five questionnaire used in our experiments, there are 44 questions in all. The score ranges of "A," "C," "E," "N," "O" are $[9, 45]$, $[8, 40]$, $[9, 45]$, $[8, 40]$, $[10, 50]$ respectively. The value of $RMSE$ shows the average difference between our prediction results and the scores of questionnaire. The smaller is the value of $RMSE$, the better is the performance of prediction model.

$$r = Cor(Y, Y') = \frac{Cov(Y, Y')}{\sqrt{Var(Y)Var(Y')}} \tag{11}$$

$$RMSE = \sqrt{\frac{\sum_{i=1}^{n}(y_i - y_i')^2}{n}} \tag{12}$$

**Table 1 The comparison of prediction results in the Pearson correlation coefficient.**

| Attributes | $r_a$ | $r_c$ | $r_e$ | $r_n$ | $r_o$ |
|---|---|---|---|---|---|
| Attribute 1 (Original) | 0.1012 | 0.1849 | 0.1044 | 0.0832 | 0.181 |
| Attribute 2 (PCA) | 0.1106 | 0.2166 | 0.1049 | 0.1235 | 0.1871 |
| Attribute 3 (Stepwise) | 0.1223 | 0.2639 | 0.1698 | 0.1298 | 0.2246 |
| Attribute 4 (Lasso) | 0.1209 | 0.2068 | 0.0788 | 0.0934 | 0.1136 |
| **Attributes SAE** | **0.2583** | **0.4001** | **0.3503** | **0.3245** | **0.4238** |

**Table 2 The comparison of prediction results in RMSE.**

| Attributes | $RMSE_a$ | $RMSE_c$ | $RMSE_e$ | $RMSE_n$ | $RMSE_o$ |
|---|---|---|---|---|---|
| Attribute 1 (Original) | 5.6538 | 6.1335 | 4.9197 | 6.5591 | 7.0195 |
| Attribute 2 (PCA) | 5.1628 | 5.6181 | 5.6781 | 5.9426 | 6.4579 |
| Attribute 3 (Stepwise) | 4.8421 | 5.3495 | 5.276 | 5.6904 | 6.1079 |
| Attribute 4 (Lasso) | 5.8976 | 6.7471 | 6.4940 | 5.4241 | 6.0938 |
| **Attributes SAE** | **4.7753** | **5.339** | **4.8043** | **5.6188** | **5.1587** |

**Table 3 The comparison of dimensionality of different feature vector.**

| Attributes | $D_a$ | $D_c$ | $D_e$ | $D_n$ | $D_o$ |
|---|---|---|---|---|---|
| Attribute 1 (Original) | 704 | 704 | 704 | 704 | 704 |
| Attribute 2 (PCA) | 250 | 203 | 250 | 310 | 250 |
| Attribute 3 (Stepwise) | 47 | 32 | 56 | 47 | 32 |
| Attribute SAE | 400 | 400 | 300 | 400 | 300 |

## Experiment results

In comparison experiments, we utilized five different kinds of attributes to train and build the personality prediction model. The five kinds of attributes include the attributes selected by artificial statistical method without feature selection (denoted by Attribute 1), the attributes selected from Attribute 1 by Principal Component Analysis (PCA) (*Dunteman, 1989*) (denoted by Attribute 2), the attributes selected from Attribute 1 by Stepwise (denoted by Attribute 3), the attributes selected from Attribute 1 by Lasso (denoted by Attribute 4) and linguistic behavior feature vector obtained based on Stacked Antoencoders (SAE) (denoted by Attribute SAE). PCA is a kind of unsupervised feature dimension reduction method, and Stepwise is usually used as a kind of supervised feature selection method. LASSO is a regression analysis method which also perform feature selection. For different kinds of attributes, the personality prediction models are all built by a linear regression algorithm. In order to obtain the stable model and prevent occurrence of overfitting for each factor of personality, we use 10-fold cross validation and run over 10 randomized experiments. Finally, the mean of 10 randomized experiments' results is recorded as the final prediction result. The comparison of prediction results of five personality **factors** using three kinds of attributes are shown in Tables 1 and 2. Table 3 shows the dimensionality of different kinds of feature vectors. The letters in subscript "a," "c," "e," "n," "o" indicate different personality factors respectively.

# DISCUSSION

This study explore the relevance between users' personality and their social network behaviors. The feature learning models are built to perform an unsupervised extraction of the representations of social network linguistic behaviors. Compared with the attributes obtained by some supervised behavior feature extraction methods, the LRFV is more objective, efficient, comprehensive and universal. In addition, based on LRFV, the accuracy of the personality prediction model could be improved.

## The performance of personality prediction model

The results in Tables 1 and 2 show that the linguistic behavior feature vectors learned through Stacked Autoencoders perform better than other attributes in both the Pearson correlation coefficient and RMSE. When using Attribute SAE, the Pearson correlation coefficients $r_e = 0.2583$, which represent a small correlation. For"E," "N," "C" and "O," $r_e = 0.3503$, $r_n = 0.3245$, $r_c = 0.4001$ and $r_o = 0.4238$, which means that the prediction results of "E," "N," "C" and "O" correlate with the results of questionnaire moderately. It is concluded that personality prediction based on the linguistic behavior in social network is feasible. Besides, the traits of conscientiousness and openness could be reflected in the network linguistic behavior more obviously.

Compared with other feature extraction methods, our proposed method performs better. When using the original feature vector (Attributes 1), the prediction results $r$ are all less than 0.2. When using another kind of unsupervised feature dimension reduction method (Attributes 2), except for "C," others are also less than 0.2. Attributes 3, which is obtained by using a kind of supervised feature selection method, the prediction results $r$ are also not ideal. Similarly, considering *RMSE* of every personality traits, the prediction model also obtain better results based on the linguistic behavior feature vectors.

Besides, we compared the time and memory consuming of prediction when using SAE and PCA to reduce the dimensionality of features respectively in Table 4. The experiments were conducted on a DELL desktop with an Intel Core 3.30 GHz CPU and 12G memories. The average time consuming denotes the average time cost for predicting one personality factor of one sample. The average memory consuming denotes the memory usage percentage when running the prediction model. Although PCA performed better in the time and memory consuming, the prediction results of linguistic behavior feature vectors were outstanding. Usually, the high-powered computing server could offset the deficiency of time and memory consuming.

## Parameters selection
### Activation function

There are many kinds of activation function in neural network, such as Sigmoid, Tanh, Softmax, Softplus, ReLU and Linear. Among them, Sigmoid and Tanh are used commonly. In experiment, we utilized both of them to construct the feature learning model, and the comparative results (Table 5) showed that when using Sigmoid as activation function of hidden layers, the prediction results are a bit better.

**Table 4  The comparison of time and memory consuming of different feature vector.**

| Attributes | Average time consuming | Average memory consuming |
|---|---|---|
| Attribute 2 (PCA) | 3ms | 56% |
| Attributes SAE | 12ms | 81% |

**Table 5  The comparison of prediction results when using different activation function.**

| Activation function | $r_a$ | $r_c$ | $r_e$ | $r_n$ | $r_o$ |
|---|---|---|---|---|---|
| Sigmoid | 0.2583 | 0.4001 | 0.3503 | 0.3245 | 0.4238 |
| Tanh | 0.2207 | 0.3338 | 0.3216 | 0.2696 | 0.3503 |

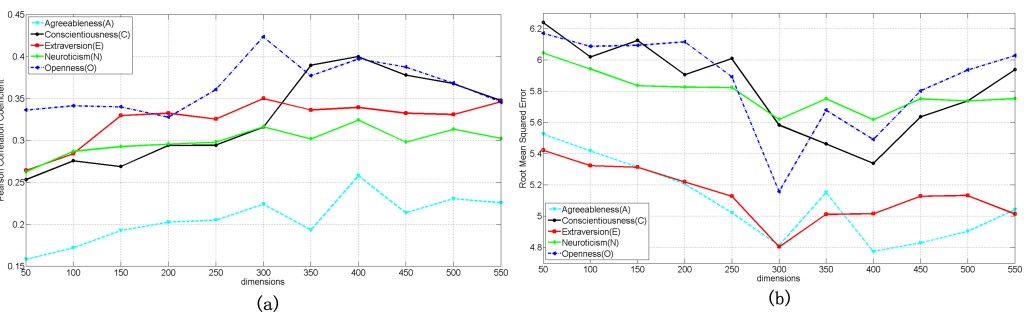

(a)                                  (b)

**Figure 4  The comparison of prediction results using linguistic feature vectors with different dimensionality.** (A) The comparison of $r$. (B) The comparison of *RMSE*.

### *The dimensionality of linguistic behavior feature vector*

For each personality trait, the dimensionality of linguistic behavior feature vector is set according to the optimal result of prediction model obtained from repeated experiments, and the comparison of $r$ and *RMSE* when using linguistic behavior feature vectors with different dimensionality are presented in Figs. 4A and 4B, respectively. The Pearson correlation coefficient reflects the correlation degree between two variables. If the change tendencies of two variables are more similar, the correlation coefficient is higher. Root Mean Square Error reflects the bias between the real value and prediction value. For a dataset, the Pearson correlation coefficient and Root Mean Square Error may not be direct ratio. In practical applications, the trend of the psychological changes is more necessary. So, when adjusting the optimal parameters, we give priority to Pearson correlation coefficient. For "A," "C" and "N," prediction models perform better when the dimensionality of feature vector is 400. For "E" and "O," we could obtain the better results when the dimensionality of feature vector is 300.

### Differences in modeling performance across personality traits

Through analysing the results of experiments, we summarize that Agreeableness correlate with users' social network linguistic behaviors relative weakly than the other personality traits. The correlation between openness and users' social network linguistic behaviors is highest of all. We could identify whether the users own higher scores in openness or not through their blogs published in social network platform, most likely because the

people with high scores in openness usually prefer to publicly express their thoughts and feelings. Similarly, conscientiousness is moderately correlate with social network linguistic behaviors. And for conscientiousness, there are significant differences of using the words belonging to the categories of family, positive emotion and so on.

## The future work

In this paper, we has carried on some preliminary study to explore the feasibility of using deep learning algorithm to construct linguistic feature learning model. More work will be conducted further. Millions of users' social media are being downloaded. In feature extraction, the massive data will be used to train the unsupervised feature learning model. Besides, a new round of user experiment is progressing. We would obtain a new set of labeled data to improve our personality prediction method. The study is of great significance. It could provide new quantitative and analytical methods for the social media data, and a new perspective for real-time assessment of Internet users' mental health.

# CONCLUSIONS

In this paper, we utilized a deep learning algorithm to investigate the correlations between users' personality traits and their social network linguistic behaviors. Firstly, the linguistic behavior feature vectors extracted unsupervised using Stacked Autoencoders models actively. Then, the personality prediction models are built based on the linguistic behavior feature vectors by linear regression algorithm. Our comparison experiments are conducted on five different kinds of attributes, and the results show that the linguistic behavior feature vectors could improve the performance of personality prediction models.

## Funding

The authors received support from the Young Talent Research Fund (Y4CX103005), National Basic Research Program of China (2014CB744600), and CAS Strategic Priority Research Program (XDA06030800). The funders had no role in study design, data collection and analysis, decision to publish, or preparation of the manuscript.

## Grant Disclosures

The following grant information was disclosed by the authors:
Young Talent Research Fund: Y4CX103005.
National Basic Research Program of China: 2014CB744600.
CAS Strategic Priority Research Program: XDA06030800.

## Competing Interests

The authors declare there are no competing interests.

## Author Contributions

- Xiaoqian Liu conceived and designed the experiments, performed the experiments, analyzed the data, contributed reagents/materials/analysis tools, wrote the paper, prepared figures and/or tables, performed the computation work.

- Tingshao Zhu conceived and designed the experiments, contributed reagents/materials/analysis tools, performed the computation work, reviewed drafts of the paper.

## Data Availability

The following information was supplied regarding data availability.

The raw data has been supplied as Data S1.

## Supplemental Information

Supplemental information for this article can be found online at http://dx.doi.org/10.7717/peerj-cs.81#supplemental-information.

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
