# Peer review of "Deep learning for constructing microblog behavior representation to identify social media user's personality"

_PeerJ Computer Science, doi:10.7717/peerj-cs.81_

## Round 0.1 · original submission · Minor Revisions

· Academic Editor

Minor Revisions

Please respond to all the reviewer comments. Please also note the comments that you should improve the English language.

Reviewer 1 ·

Basic reporting

The paper is basically well-organized and well-written. However, there are several grammar mistakes I noticed that lead to unclarity of readers' understanding. They are listed as follows:
1. In abstract, "which could unsupervised extract Linguistic Representation Feature Vector (LRFV) from text published on Sina Micro-blog actively." where "unsupervised" should be "unsupervisedly", or the authors can change the structure of the sentence. Same goes to others.
2."Collecting volunteers’ IDs of Sina Micro-blog, we crawled their micro 119
blog data through Sina Micro-blog API. The micro-blog data collected consists the users’ all blogs 120 and their basic status information, such as..." Obviously "crawled" should be replaced with words like "obtained"and "consists" should be "is consist of" since consist is an intransitive verb.
3. In 5.1 "there are weak correlation..." should be uncorrected use.

I think you need to ask a native for revision of English.

Experimental design

The authors describe their methods of study pretty clearly and successfully. The key point of this study lies largely on how good is the "dictionary". I actually looked into the "LIWC" 1 years ago. This software is based on English key word counting process and neglects so many important grammars. Thus the correct rate of it is just so so.

I noticed your decryption of your "dictionary". My question is how precise your algorithm performs, and in what way is yours different with previous methods?

Also I would like to know why you choose linear regression for prediction.

Validity of the findings

How you would consider the performance of the proposed algorithm valid? I have seen enough description on this part.

Additional comments

The paper is basically well-organized and well-written with interesting results and findings. Still it need some kind of revision. Details can be found in separated parts.

Reviewer 2 ·

Basic reporting

The notation might be ambiguous: author first use the word 'dimension' for the 5 traits (line 208), then use 'dimension' to represent the number of the features for prediction. It would be better if two term could be distinguished.

Figure 3 is only a deep architecture of stacked autoencoders. However, authors should include the exact structure that used in their study, instead of a general structure (for example, number of units in each layer, and number of layers). Or at least a table of the network used should be represented. You can find examples in the famous Alex net paper.

Sample size is relatively small in this setting (1552). With this sample size, deep learning is not so necessary. It is better to explain how and why deep learning works in such small size, compared to other algorithms.

Experimental design

1. In line 215-216, Authors trained five personality prediction models based on linear regression algorithm using corresponding linguistic behavior feature vectors and labels. Does these mean the five linear models are trained separately? This might lose the information between five dimensions.

2. Line 275 mentions sigmoid function is used as activation function of hidden layers. There are many choice for the nonlinearity units in the network. Is there any experiment with other units, or are there any literatures suggest the preference?

3. Line 124 mentions: If the total number of one’s micro-blogs is more than 500, this volunteer is a valid sample. This is reasonable, but might cause bias in the experiment. Authors might be more careful of this step and make more discussion for the potential outcome of this step.

4. The dimensionality of linguistic behavior feature vector is set according to the optimal result of prediction model. Did authors try first generate high dimensional data, then use some statistical method for feature selection, e.g. lasso?

Validity of the findings

1. In the section 'Evaluation measures', authors did not mention explicitly what is "prediction". Does the data split into training and testing data? Did testing data used in feature extraction?

2. Authors study the relationship of the number of features and the prediction performance. There is an interesting pattern in figure 4b that 300 is a local minimum, while does not shown in the 4a. Authors might add more discussion of why that happens.

Reviewer 3 ·

Basic reporting

No Comments

Experimental design

No Comments

Validity of the findings

No Comments

Additional comments

In this paper, the authors use deep learning methods to study the social media user's personality problem. Both the topic and technique is quite new.
My major comments are as follows,

1. For data collection, linguistic behavior is extracted based on the FFT methods. By applying FFT, we always assume the signal is stationary. However, whether micro-blog data is stationary signal or not is still highly questionable. Since micro-blog data does not only depend on user's thoughts, ideas and feelings. It is also strongly influenced by public events, like recently death of college student Zexi Wei raises questions on Baidu's ethics, which typical make the data non-stationary.

2. In method, Stacked Autoencoders is used as a deep learning method for dimension reduction, which is an unsupervised method. The author mentioned that compared with supervised methods unsupervised method is more objective, efficient, comprehensive and universal. However, normally supervised dimension reduction method would achieve a better performance than unsupervised methods. It is better to make a further discussion and compare different unsupervised and supervised dimension reduction methods in result.

3. In result, the authors use different parameters for the demission of feature vector. What kind of principle has been used here for parameter selection? If the authors select the parameters corresponding to the best result, it would be unfair to the other methods in the comparison and also unable to be used in practical. And also, how many components are used in the PCA? Again, what principle has been used for parameter selection in PCA? Furthermore, the introduction of the SAE is not sufficient? There is no definition of function f and g. What does \theta and \theta' stand for in Eq. (4) and Eq. (5)? How to determine the demission of x'? How to training the W, b, W', b in Eq. (4) and (5)? As the authors mentioned in line 200, the number of layer would be decided according to the optimal value of many experiments? Which kind of experiments will be done? Is it time consuming or did we need additional data in these experiments. Finally, it should also mentioned the time and memory consuming for the method of Stacked Autoencoders in this application, and make a comparison with the others, like PCA. In all, a lot of technique detail has not been described, which makes the readers unable to get a comprehensive understanding of the proposed method.

Minor questions:
1. Does all the users have their micro-blogs account 3 years ago?
2. Eq. (9) and (10) is not so strict, Eq. (9) is in a vector form and Eq. (10) is in a scalar form.

---

## Round 0.2 · Minor Revisions

· Academic Editor

Minor Revisions

Some revisions are still needed.

Reviewer 2 ·

Basic reporting

No further comments.

Experimental design

No further comments.

Validity of the findings

No further comments.

Additional comments

After revision, this paper meets the standard for publishing. I would suggest accept this article.

Reviewer 3 ·

Basic reporting

No Comments

Experimental design

No Comments

Validity of the findings

No Comments

Additional comments

I’m the third reviewer for the previous version of the manuscript.
It’s great honor for me here to be the reviewer again for the revision. But I still should say, the authors did not answer my questions well.

My first question is why FFT is used for feature extraction. In the reply the authors showed me more detail about the result after FFT. However the authors still did not make any explanation about the reason of using FFT.

The second question about supervised and unsupervised methods. The authors mentioned that, "for supervised methods, the directivity of dimension reduction results is stronger"; and the reason for using unsupervised method is that "there are some limitations of obtaining labeled data". In fact, after the dimension reduction, the label information is still needed for regression. So logically, the label information must be known for the proposed method. Since this, I did not find any reason here for using unsupervised deep learning method in dimension reduction instead of some supervised methods.

The last question is about the principle of parameter selection. Since there are so many parameters need to be determined in the proposed method, parameter selection would be key important for the success of the method. Here, the authors determine the value of the parameters by the optimal prediction results. This principle is the most direct way for parameter selection. However it is easy to lead to the parameters overfitting to the dataset, which makes the results harder to believe. Cross validation is an alternative approach, which will leads to more computation. But at least it should be done on selection of the most important parameters.

---

## Round 0.3 · accepted · Accept

· Academic Editor

Accept

Based on the comments, I can suggest the acceptance of this paper.

Reviewer 3 ·

Basic reporting

No Comments

Experimental design

No Comments

Validity of the findings

No Comments

Additional comments

In the revision, I see a great effect has been made by the authors. All the questions I asked has been answered well. The quality of the manuscript has been improved a lot. So I suggest to accept it for PeerJ.